# Tracking extinction risk trends and patterns in a mega-diverse country: A Red List Index for birds in Colombia

**Luis Miguel Renjifo**[1]*, **Angela María Amaya-Villarreal**[1], **Stuart H. M. Butchart**[2]

**1** Department of Ecology and Territory, School of Environmental and Rural Studies, Pontificia Universidad Javeriana, Bogotá, D.C., Colombia, **2** Department of Zoology, BirdLife International, University of Cambridge, Cambridge, England, United Kingdom

* lmrenjifo@javeriana.edu.co

**Data Availability Statement:** All relevant data are within the manuscript and its Supporting Information files. Data regarding species distributions in regions, ecosystems, species

## Abstract

Monitoring trends in the extinction risk of species is important for tracking conservation effectiveness. The Red List index (RLI) reflects changes in aggregate extinction risk for sets of species over time (a value of zero means that all species are extinct, a value of one means that all species are categorized as Least Concern). We calculated the first national RLI for birds in Colombia for the period 2002–2016, and disaggregated indices by ecosystems, regions, and species groups. Overall, the status of birds in Colombia has moderately deteriorated during 2002–2016, declining by 0.0000714% per year (the global RLI for birds declined by 0.0297% per year). High Andean forest, paramo, and freshwater are the ecosystems in worst condition. The two regions with the greatest avian diversity contrasted: the Andes has the lowest RLI, and the Amazon the highest. Among species groups, gamebirds, parrots, large frugivores, and forest raptors are the most threatened. Habitat loss from expansion of illicit crops and population declines from hunting were the most important threats. Agricultural expansion, invasive alien animal species, illegal logging and illegal mining are significant threats for some species. Tracking species' extinction risk is important in a country with the highest bird species richness in the world, dynamic spatial patterns of habitat loss, and high levels of endemism. Recent developments provide reasons for both hope and despair. In 2016, a peace agreement ended 50 years of armed conflict. New opportunities for biodiversity conservation, local development based on bird-watching tourism, and advancement in scientific knowledge of birds now occur alongside dramatic increases in deforestation. These new conservation opportunities and challenges provide strong motivation to take advantage of the fact that the overall risk of extinction of birds in Colombia is still relatively low and stable. Effective action is urgently needed while there still is the opportunity to prevent extinctions and safeguard species, particularly those in higher risk categories.

groups were obtained from published literature and paper´s authors and are included as supplementary information

**Funding:** LMR. This work was made possible with funding Vice-presidency for research of Pontificia Universidad Javeriana (VRI Project number 7284).

**Competing interests:** The authors have declared that no competing interests exist.

## Introduction

Human activities have become a force of global significance, causing disruption to climate, landscape-scale habitat loss, pollution, and overexploitation of biological resources [1,2,3,4]. As the human population has grown in numbers and consumption, the growing proportion of natural resources that consumes has caused habitat loss and species extinction [5,6]. Habitat loss and unsustainable levels of hunting associated with armed conflict and illegal drug-cultivation are a particular threat to biodiversity in some tropical regions [e.g. 7]. Extinctions and declines in native species' abundance undermine the capacity of ecosystems to provide ecosystem services such as provision of food, materials, and drinking water, with direct consequences for human health [2,5]. Declines in bird populations, in particular, may reduce the delivery of ecosystem services such as pollination, seed dispersal and pest-control [8,9].

Extinction risk assessment has become an essential tool for conservation planning and action [10], with the IUCN Red List widely regarded as the global standard. The process of undertaking Red List assessments not only produces an evaluation of the conservation status of species, but also involves compiling an extraordinary wealth of data on which such assessments are based [11]. Red Data Books and Lists provide support for a variety of actions such definition of conservation priorities, allocation of economic resources, environmental education and raising awareness of the biodiversity crisis [11,12].

Periodic risk assessment allows for monitoring trends over time, detecting changes in causes of threat and their intensity, and providing insights into the effectiveness of conservation policies [11,12]. To achieve this and allow meaningful comparison of repeated extinction risk assessments, the Red List Index (RLI) was developed [13,14,15]. RLIs illustrate the relative rate at which groups of species change in their overall level of extinction risk. They are calculated from the number of species in each Red List category and genuine changes between assessments (i.e. those resulting from improvements or deteriorations in the status of species). Those changes due to improved knowledge or taxonomic status are excluded [13, 15]. RLIs have been used to track global and regional trends in the extinction risk of species [13,16,17,18], and for subsets of species or drivers of trends relevant to different policy targets, such as the impact of invasive species or trends in pollinator species [e.g. 19,20,21], as well as to track institutional impact on conservation [22]. On a global scale, the index is available for birds, mammals, amphibians, corals, cycads and conifers [e.g. 23], with baseline data points available for a number of other vertebrates, invertebrate and plant groups, including some derived from a sampled approach [e.g. 24,25]. The latter entails assessing a random selection of species from across an entire taxonomic group, allowing for the assessment of trends for large and poorly known taxonomic groups when the sampled species are reassessed [26].

Most conservation policies are implemented at the national scale, where conservation resources are also typically allocated. The RLI can be applied at the national scale through disaggregation of the global index [27], by weighting each species according to the proportion of its range (a surrogate for population) within the country, to account for national responsibility for the conservation of each species [28]. National RLIs can also be produced from repeated assessments of extinction risk at the national scale through national Red Lists, with examples published mainly for vertebrates, but also for invertebrates, plants and lichens, spanning terrestrial, freshwater and marine realms, and temperate and tropical countries such as Australia, China, Denmark, Ecuador, Finland, Italy, Paraguay, Spain, Sweden, and Venezuela (reviewed in [29], see also [16,30]). Such national RLIs tend to be more sensitive in detecting national biodiversity trends because larger numbers of species change Red List category between assessments when extinction risk is assessed at national rather than global scales [18]. However, they

may also reflect changes in the local status of species that are of minor significance at a global scale [31].

Birds are the best-known class of organisms, and are useful indicators of broader biodiversity trends [32]. According to BirdLife International, and the South American Classification Committee, Colombia has the most diverse avifauna in the world, with over 1900 species, 4.4% of them endemic to the country [33,34,35,36]. In addition, Colombia contains global biodiversity hotspots (e.g., tropical Andes and Chocó/Darién, [37]), and regions with high levels of endemism and species richness [38,39]. For these reasons, bird conservation in Colombia is of global significance. While national red list assessments indicate that the proportion of threatened bird species has increased from 6.4% in 2002 to 8.1% in 2016 [40,41,42,43], this variation is partly a result of changes in knowledge and taxonomy. We calculated the first Red List Index for birds in Colombia to determine the genuine change in national extinction risk of birds in Colombia over this period, compare it with global trends, and explore patterns across ecosystems, regions and groups of species of conservation concern.

## Methods

In order to calculate an RLI for the country we followed Butchart *et al*. [15]. To obtain a complete list of bird species found in Colombia, we used Avendaño *et al*. [35], but excluded passage migrants (transients), vagrants, introduced species, and species for which their occurrence within the country is unconfirmed according to Avendaño *et al*. [35] and the authors' knowledge. We used the two most recent assessments of extinction risk for birds for the country [40 and 41 and 42, hereafter summarized as 42]. These two assessments applied the IUCN Red List categories and criteria [44] and followed the IUCN guidelines for their regional/national application [45]. The categories considered were extinct (EX), critically endangered (CR), endangered (EN), vulnerable (VU), near threatened (NT) and least concern (LC). Nine species evaluated as data deficient in 2016 were excluded, following Butchart et al. [15]. This produced a list of 1718 species for which we calculated the RLI. For each of these, we followed the approach outlined in Butchart et al. [15] and assessed if any had undergone genuine improvements or deteriorations in status of sufficient magnitude such that they would have qualified for a lower or higher Red List category in 2002, using information in Renjifo *et al*. [40,42]. We excluded changes in category between assessments that resulted from better knowledge, differences in methodology (for example different interpretation of information or different process of assessment due regional adjustments) or taxonomy (newly described species, split or lumped taxa) [13,15]. For genuine changes, we documented which parameters changed, triggering which criteria and resulting in which category shifts, as well as the drivers (threats or conservation actions).

We calculated the RLI for each time point, assigning "equal steps" weights for each IUCN Red List category (0 for LC, 1 for NT, 2 for VU, 3 for EN, 4 for CR and 5 for EX). The number of species in each category was multiplied by these weights and the products summed. The total was divided by the maximum possible value (# of non-DD species multiplied by 5) and subtracted from one to give the RLI. An RLI value of one results if all species are classified as least concern; an RLI value of zero results if all species are classified as extinct. In other words, the index declines as the status of species deteriorates, to be consistent with most other biodiversity indicators.

We calculated disaggregated RLIs for species in different regions, ecosystems and groups of species of conservation concern, assigning species to these using the information found in Hilty & Brown [46], Ayerbe [47], Schulenberg [48], and expert knowledge of authors. The occurrence of some marine and migratory species within some regions of the country was

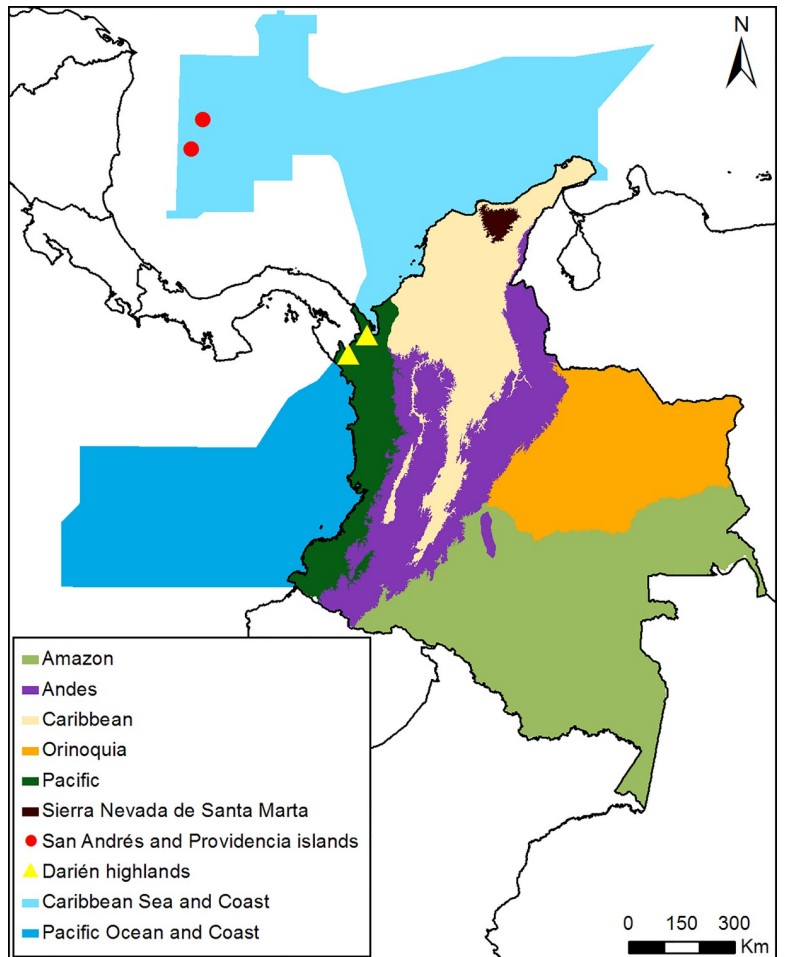

**Fig 1. Regions of Colombia used for disaggregated RLIs.** San Andrés and Providencia Islands and Darién Highlands are represented with symbols because they are too small to be seen at the scale of the map.

determined following consultation with experts (see acknowledgments). Regions were defined as: Pacific (including all lowlands on the Pacific side of the Andes that encompass both Darién and Chocó biogeographic regions, including dry forest within Patía valley); Pacific coast and ocean (coastline, islands and open sea); Andes (all three main mountain ranges and Perijá); Caribbean (continental region, including Magdalena and Cauca valleys); Caribbean coast and ocean (coastline, coastal islands and open sea); Sierra Nevada de Santa Marta (SNSM, including foothills); Darién Highlands; San Andrés and Providencia islands; Amazon; and Orinoquia (Fig 1). Ecosystems were defined as: lowland rainforest, lowland dry forest (including dry scrub, and desert), Sub-Andean forest (both rain and dry), High Andean forest (both rain and dry), paramo, mangrove forest, savanna, freshwater ecosystems (wetlands and rivers), coastal and pelagic waters. We lumped together species of rain and dry Sub-Andean and High Andean forest because current knowledge precludes a consistent classification of species associated with dry mountain forests. Lowland was defined as <1000 m, Sub-Andean as 1000–2500 m, and High Andean as >2500 m. For species in the Darién mountains, the cut-off between lowland and Sub-Andean was taken as 800 m, because on small isolated mountains the upper limit of lowland rain forest is lower than on major ranges due to the "Massenerhebung" effect [49]. Species groups of conservation concern were defined as: gamebirds (Tinamiformes,

Anseriformes, Galliformes, Columbiformes), diurnal raptors (Accipitriformes and Falconiformes), parrots (Psittaciformes), terrestrial forest insectivores (Conopophagidae, Grallaridae, Rhinocryptidae, Formicariidae, Furnariidae, and some Cuculidae, Thamnophilidae, Tyrannidae, Troglodytidae, Polioptilidae and Parulidae), large frugivores (species that consume ripe or unripe fruit on a regular basis and weigh more than 100 g: Tinamidae, Cracidae, Odontophoridae, some Columbidae, Steatornithidae, Psophiidae, some Trogonidae, some Momotidae, Ramphastidae, some Picidae, some Psittacidae, some Cotingidae, some Corvidae, some Turdidae, some Thraupidae, some Icteridae), and forest-dependent raptors (some Accipitridae, and some Falconidae), while control groups were nocturnal raptors (Strigiformes), suboscine and oscines passerines, and hummingbirds (Trochilidae) (See S2 Table). We included terrestrial forest insectivores, large frugivores, and forest-dependent raptors because they have been found to be consistently prone to extinction due to habitat fragmentation [e.g. 50,51,52].

We compared the number of threatened (CR, EN, VU), near threatened (NT) and least concern (LC) species on the global IUCN Red List [53], with the numbers of species in the same categories in the Colombian national Red List [41,42]. Then, we compared the number of species in each category on the national list with the number of Colombian species in each category on the global IUCN Red List [53]. We conducted G-tests to evaluate statistical significance, and used a Fisher Exact test to compare the number of extinct species in the Global Red List with the number of extinct species in the national Red List.

## Results

A total of 1909 species have been recorded within Colombia [35], of which 1727 are either residents (species that spend all year in Colombia) or seasonal migrants (i.e., temperate and intratropical migrants). Among them, one endemic species is extinct (Colombian Grebe *Podiceps andinus*). Comparing the Colombian National Red List with the Global IUCN Red List we found that there is a significantly lower proportion of nationally threatened species (8.1%) than globally threatened (13.6%) (G-test = 219.463, df = 4, p <<0.0001), and a higher proportion of species that have gone extinct since 1500 at global than at national level (Fisher´s Exact test, p<<0.001). Comparing the Colombian National Red List with the Colombian species in the global IUCN Red List (i.e. species present in Colombia assessed by the IUCN Red List), we found a higher proportion of species threatened in the national assessment (8.1%) than in the global assessment (6.7%) (Table 1), (G-test = 51.623, df = 5, p<<0.0001). This was expected because species are more likely to qualify as threatened when extinction risk is assessed at a national rather than global scale [18].

We found that most of the changes in species' red list categories between 2002 and 2016, a total of 70 (81.4% of species with changes), were due to better knowledge of species distributions, populations, trends, etc., or changes in the way in which the red list categories were applied, or improved analytical methods (following 41,42,45). A smaller proportion of category changes (18.6%) was due to genuine improvement or deterioration in species' conservation status. From 1718 extant non-DD species, 16 qualified for lower (3) or higher (13) categories of extinction risk owing to genuine changes (Table 2). These category changes drive trends in the RLI, although the full set of species and their current red list categories are used to determine the 2016 RLI value [13,15]. The causes of improvement in conservation status differed for the three species. One of them recovered due to land abandonment and subsequent habitat recovery as people moved to cities [54,55,56]; another experienced an improvement due to a reduction in the rate of habitat loss as illegal cultivation of coca shifted among regions [41,57]; and a third benefited from conservation actions. For the uplisted species, habitat loss from expansion of illicit crops and population declines from hunting were the two

**Table 1. Bird species (percentage in parenthesis) in each IUCN Red List category in the world and in Colombia.** For Colombia, we excluded vagrants, passage migrants, introduced species, and species of uncertain occurrence. The percentage of threatened species was calculated in relation to the total number of extant species. EX: extinct, EW: extinct in the wild, CR(PEW): critically endangered (possibly extinct in the wild), CR(PE): critically endangered (possibly extinct), CR: critically endangered, EN: endangered, VU: vulnerable, NT: near threatened, LC: least concern, DD: data deficient. Global totals are based on the 2018 IUCN Red List. Totals for Colombian species on the global IUCN Red List include passage migrants.

| Red List Category | Global IUCN Red List | Colombian species on global IUCN Red List | Colombia National Red List |
|---|---|---|---|
| EX | 156 (1.4%) | 1 (0%) | 1 (0.1%) |
| EW | 5 (0%) | 0 (0%) | 0 (0%) |
| CR(PEW) | 1 (0%) | 0 (0%) | 0 (0%) |
| CR(PE) | 21 (0.2%) | 0 (0%) | 2 (0.1%) |
| CR | 202 (1.8%) | 14 (0.8%) | 14 (0.8%) |
| EN | 469 (4.2%) | 36 (1.9%) | 56 (3.2%) |
| VU | 799 (7.2%) | 76 (4.0%) | 68 (3.9%) |
| NT | 1012 (9.1%) | 105 (5.6%) | 27 (1.6%) |
| LC | 8405 (75.5%) | 1641 (87%) | 1550 (89.8%) |
| DD | 56 (0.5%) | 5 (0.3%) | 9 (0.5%) |
| TOTAL | 11126 | 1878 | 1727 |
| TOTAL THREATENED | 1492 (13.6%) | 126 (6.7%) | 140 (8.1%) |

**Table 2. Genuine changes suffered by species during 2002–2016 period.** Endemic species are marked with asterisk*.

| Family | Common name | Species | Change | Support |
|---|---|---|---|---|
| Anatidae | Orinoco Goose | *Oressochen jubatus* | uplist | The species qualified for uplisting from NT to VU under criterion A2 due to an acceleration in the rate of habitat loss in the llanos region (Orinoquia), owing to intensifying habitat destruction for agriculture and extensive livestock farming. By 2002 the rate of decline was projected to be below 30% in 3 generations but during 2002–2016 exceeded 30% because of increases in the rate of loss of forest (which is used by the species when nesting) and hunting. Also, the population size decreased and crossed the threshold under criterion C2 from 25000 individuals by 2002 to 5000–10000 individuals by 2016. |
| Cracidae | Great Curassow | *Crax rubra* | uplist | The species qualified for uplisting from NT to VU under criterion A2 owing to increased hunting and deforestation in its distribution range due to illegal coca crops, illegal gold and platinum mining, and illegal timber extraction. By 2002 the rate of decline was projected to be below 30% in 3 generations but during 2002–2016 exceeded it because of an increased rate of forest loss, degradation and hunting. |
| Cracidae | Yellow-knobbed Curassow | *Crax daubentoni* | uplist | The species qualified for uplisting from VU to EN because the population size is suspected to have fallen below 2500 individuals by 2016, resulting in uplisting under criterion C2. Also, the species crossed the threshold under criterion A2 from VU to EN because the rate of population decline was suspected to have exceeded 50% over last three generations (and predicted to continue in the future). All these changes are due to intensifying of deforestation and fragmentation especially for illicit crops and presumably an increase in hunting. |
| Cracidae | Helmeted Curassow | *Pauxi pauxi* | uplist | The species qualified for uplisting from VU to EN under criterion A2 due to a decline in the population driven by habitat loss from illicit crops and concomitant hunting, especially in Catatumbo and Magdalena medio regions. By 2002 the rate of population decline was projected to be 30% over next 3 generations but during 2002–2016 it increased to ≥50% because of an increase in deforestation rate and hunting. |
| Podicipedidae | Northern Silvery Grebe | *Podiceps juninensis* | uplist | The species qualified for uplisting from EN to CR under criterion B2 because it disappeared from La Cocha lagoon, presumably because of hunting and degradation of habitat. This decreased its AOO below 10km². The main drivers of the lagoon's degradation are eutrophication and the introduction of invasive trout. |
| Trochilidae | Black Inca | *Coeligena prunellei** | downlist | The species qualified for downlisting from EN to NT due a recovery of its habitat. This hummingbird uses both mature forest and regenerating areas. Some conservation actions have taken place since 2005 including the active recuperation of oak forest (*Quercus humboldtii*) in Boyacá and the protection of the species in Santander. |

(*Continued*)

**Table 2.** (Continued)

| Family | Common name | Species | Change | Support |
|---|---|---|---|---|
| Capitonidae | Orange-fronted Barbet | *Capito squamatus* | uplist | The species qualified for uplisting from NT to VU under criteria A4 and B1. By 2002, this species was below the threshold of population reduction over three generations spanning the past and future. However, the species lost 15% of its habitat during 2001–2010, so is projected to lose $\geq$30% of its population over next three generations. The species also met the threshold of VU for criterion B1 due to the intensification of deforestation within its distribution (Nariño department, southwest Colombia) during the last decade. Deforestation in this region is due to the expansion of illicit crops, and efforts to eradicate them. |
| Rhamphastidae | Plate-billed Mountain-toucan | *Andigena laminirostris* | uplist | The species qualified for uplisting from VU to EN under criterion B1 due to the intensification of deforestation within its distribution (Nariño department, southwest Colombia). The AOO in 2002 was around 1770 km$^2$ but during the last decade it fell below 500km$^2$ because of increasing deforestation for illicit crops and efforts to eradicate them. |
| Picidae | Guayaquil Woodpecker | *Campephilus gayaquilensis* | uplist | The species qualified for uplisting from NT to VU under criterion B1+2 due to declines EOO and AOO, and under C2 due to habitat loss. During 2002–2016, there was increased destruction of its habitat (mangroves and lowland humid forest), in the south-west Pacific region (Nariño and Cauca departments). The main drivers of habitat degradation are the expansion of agricultural frontier and illicit crops of coca. |
| Falconidae | Plumbeous Forest-falcon | *Micrastur plumbeus* | uplist | The species qualified for uplisting from NT to EN under criterion C1. In 2002 the rate of population decline was below the threshold of 10% over the next three generations. However, by 2016 it was projected that the species will lose 25% of its population over next two generations, because habitat loss is occurring faster owing to intensifying of deforestation in southwest Colombia. The main drivers of habitat loss and degradation are illegal crops of coca and the use of agrochemicals to which raptors are sensitive. |
| Psittacidae | Yellow-eared Parrot | *Ognorhynchus icterotis* | downlist | The species qualified for downlisting from CR to EN under criterion C because during 2002–2016 it experienced a recovery of its population. This improvement is due to conservation actions in Tolima, Antioquia-Caldas and Meta departments including habitat protection and restoration as a consequence of government and civil campaigns. Also, the species has benefited from artificial nest boxes. The species remains in EN category instead VU or NT because some relict populations still are without protection and because each population is small (<250 mature individuals). (The discovery of new populations after 2002 also meant it no longer met the B criteria owing to improved knowledge). |
| Rhinocryptidae | Stiles's Tapaculo | *Scytalopus stilesi** | uplist | The species qualified for uplisting from VU to EN under criterion B2. This species was discovered after 2002 (described as new to science in 2005) but we retrospectively assessed it VU in 2002. During 2002–2016, the species suffered habitat loss and fragmentation, especially in the north of its distribution (Antioquia department). In this period the AOO fell below 500km$^2$, qualifying the species as EN. |
| Tyrannidae | Bearded Tachuri | *Polystictus pectoralis* | uplist | The species qualified for uplisting from NT to VU under criterion A2+3. Due to an acceleration in the rate of habitat loss in the llanos region (Orinoquia), by 2016 the rate of decline was projected to be 30% over next 3 generations. Since 2002 a rapid replacement of natural savannas by large-scale crops, cattle expansion and oil extraction infrastructure has occurred. Another important driver of habitat degradation for this species is the increased frequency of fires. |
| Cotingidae | Fiery-throated Fruiteater | *Pipreola chlorolepidota* | downlist | The species qualified for downlisting from VU to NT due to a recovery of its habitat. Under criterion A, in 2002 the rate of decline was projected to be above or equal to 30% in 3 generations because of expansion of illegal crops in Amazonian piedmont (Putumayo department). However, during 2002–2016 the rate of decline fell below 30% because the illicit crop cultivation moved to other regions, especially the Pacific region (Nariño department), allowing the recovery of habitat within the species' distribution. |
| Troglodytidae | Apolinar's Wren | *Cistothorus apolinari** | uplist | The species qualified for uplisting from EN to CR under criterion B2 because it disappeared from an important area of remaining habitat (i.e. most localities of sabana de Bogotá), and the current distribution is severely fragmented. This situation diminished its AOO below 10km$^2$ during 2002–2016. This species has its distribution fragmented because of the destruction and degradation of its habitat (including from pollution in wetlands). Other threats are hunting, depredation of nests by rats, and its low capacity to colonize new areas. Finally, this species is affected by social parasitism of Shiny Cowbird (*Molothrus bonariensis*). |
| Thraupidae | Scarlet-breasted Dacnis | *Dacnis berlepschi* | uplist | The species qualified for uplisting from VU to EN under criterion B1+2 due to the intensification of deforestation within its distribution (Nariño department, southwest Colombia). During the last decade deforestation in this region increased due to the expansion of illicit crops and efforts to eradicate them. This situation diminished its AOO below 500km$^2$. |

most important causes of deterioration in conservation status. Of the 13 uplisted species, eight were affected by expansion of illicit coca crops, seven experienced intensified hunting (one also possibly by trapping), three were impacted by invasive and domestic animals (trout, cats, rats, dogs, and Shiny Cowbirds), two by agriculture including cattle ranching, two by timber extraction, two by illegal mining or oil production, one by water pollution, and another by habitat loss from development. Most species were affected by multiple threats. Most of the species that were uplisted had restricted distributions, particularly in south-west Colombia, including both Pacific and Andes (Nariño department mainly, but also Cauca and Valle del Cauca). It is notable that the 13 uplisted species include three cracids, reflecting the disproportionate susceptibility of this group to hunting and habitat loss [43].

Integrating these genuine Red List category changes, the Red List Index showed a rather stable trend in the aggregate survival probability (the inverse of extinction risk) of Colombian birds, declining from 0.955 in 2002 to 0.954 in 2016. Disaggregation showed that birds in the Andes are most threatened (i.e. lowest RLI values), while those in Amazon and Orinoquia are least threatened (Fig 2A, S1 Table). We found that the proportion of species in different risk categories are not significantly different between the national scale and the Andes region (G-test = 1.195, df = 4, p = 0.879), but significantly different between the national scale and the Amazon, and between the national scale and Orinoquia (G-test = 92.198, df = 4, p $\ll$ 0.0001

**A**

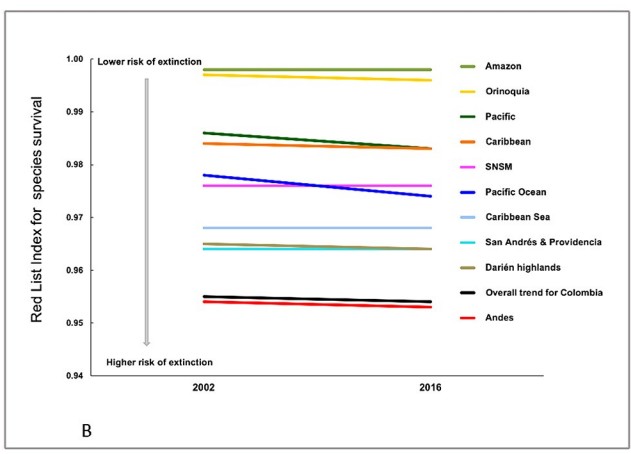

**B**

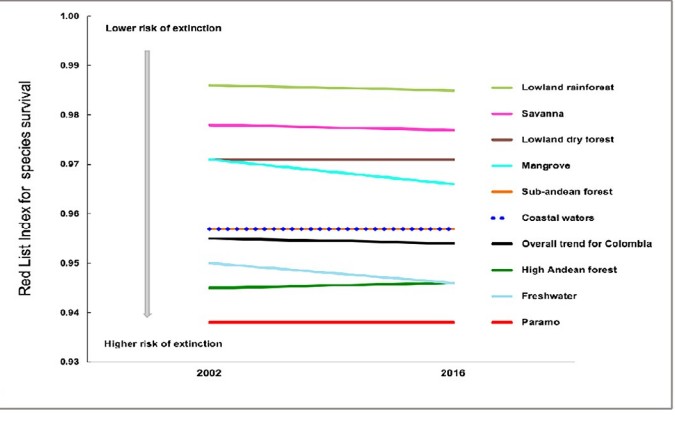

**C**

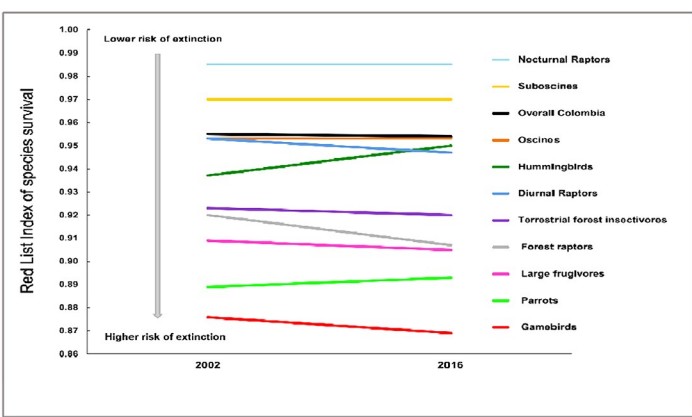

**Fig 2.** Disaggregated Red List Indices for birds in Colombia in different groups (A) regions, (B) ecosystems, and (C) groups of species of conservation interest. The overall national RLI is shown in black.

for the Amazon, G-test = 70.437, df = 4, p << 0.0001 for Orinoquia). Comparing habitat types, species inhabiting lowland rain forest are least threatened while freshwater species and species of High Andean forest and paramo have the highest extinction risk (Fig 2B, S1 Table). The small number of genuine changes mean that differences in RLI trends for different disaggregations were not substantial, but it was notable that species inhabiting mangroves and freshwater ecosystems or occurring in the Pacific region and Pacific Ocean showed the most pronounced negative trends (Fig 2A, S1 Table). In addition, gamebirds, parrots, large frugivores, terrestrial forest insectivores, and forest raptors are groups with relatively high levels of extinction risk (lower RLI values; Fig 2C, S1 Table).

## Discussion

Comparisons between national and disaggregated RLIs can highlight ecosystems and regions to focus on to prevent extinctions. The avifaunas of paramo, high Andean forest and freshwater ecosystems have RLIs below the national index (meaning that, overall, species are more threatened in these ecosystems), with freshwater species showing the greatest negative change between 2002 and 2016 (Fig 2B). Furthermore, eight out of 14 threatened freshwater species are exclusively found in the Andes, while two are shared between the Andes and lowlands. By contrast, lowland rainforest, savanna, and lowland dry forest all have high and stable RLIs, indicating that their bird species are less threatened overall (Fig 2B). The poor conservation status of freshwater birds, and especially montane freshwater species, was notable: these species appear to have been overlooked as a group of special concern.

Among regions, those with the greatest avian diversity, namely the Andes and Amazon, stand out for their contrasting situation (Fig 2A). The Andes has the lowest RLI: this region has many range-restricted species, and has experienced extensive agricultural activities and deforestation for centuries [58,59]. Consequently, the Andes have many threatened species, are the only region with an RLI below the national index, and have a very strong influence on the overall index (Fig 2A). By contrast, the Amazon has a more recent history of landscape change [58,60], and very few range-restricted species [38]. The Amazon (along with Orinoquia) still has an RLI close to one. In fact, the proportion of species in different risk categories are not significantly different between the country and the Andes region, but significantly lower in the Amazon and Orinoquia. However, there is currently very rapid and extensive habitat loss in the western parts of the Amazon, as well as in Orinoquia, due to rapid expansion of intensive cattle ranching and agriculture, oil palm plantations, and illicit coca crops [61,62,63,64]. Two other regions deserve particular attention: the Darién Highlands, and San Andrés and Providencia. Los Katíos National Park covers only part of the Darien lowlands, but no highland species are protected within it, and there is only one small terrestrial protected area on each of San Andrés and Providencia islands [65]. They have low RLI values and a high proportion of species with extremely small distributions. There are no protected areas within the Darién Highlands in Colombia, and there is very limited protection within San Andrés and Providencia. In combination, these results indicate that both montane and insular regions should be important targets of conservation action in Colombia (noting that these are also the regions that are likely be most negatively affected by climate change; [66,67]), while freshwater ecosystems also deserve greater attention.

We found that relatively few species underwent genuine deteriorations in status, with the Colombian RLI declining by 0.0000714% per year during 2002–2016. By comparison, the global RLI for birds declined by 0.0297% per year during 1988–2016 [68], again suggesting that, on average, the level of threat to birds in Colombia is worsening at a slower pace than at a global scale. The relatively high national RLI values for birds in 2016 and the relatively low rate

of decline in the RLI during 2002–2016 provide encouraging news, especially for a country with a very large avifauna. The proportion of species that qualify as threatened was lower for the national scale assessment of extinction risk in Colombia than for birds globally (Table 1), despite the fact that at a national scale species have smaller populations and distributions and therefore are more likely to meet the thresholds for threatened categories under some criteria. The difference is particularly strong for extinct species, reflecting the fact that most global extinctions have been on remote oceanic islands [69], which are poorly represented in Colombia. This may be the effect of a variety of factors. Among them, the large number of Amazonian and Orinoquian species that tend to have broad distributions in lowland Colombia, and hence tend not to qualify as threatened even at a national scale. Also, more than 50% of the land area in the country is still covered by native forest, and there are vast areas of other natural non-forest ecosystems such as savanna and paramo [70,71]. This contrasts with only 30.6% of original forest cover remaining globally [72,73,74].

The RLI is moderately sensitive, because it only reflects changes in status that are sufficient to qualify a species for a lower or higher Red List category. Many more species undergo improvements or deteriorations that are smaller in magnitude and not reflected in the index [75,76]. For example, Cauca guan (*Penelope perspicax*) has increased in population size since 2002, particularly in the Otún River basin, Yotoco and probably in Bremen and Barbas river canyon, in response to habitat protection and hunting controls [41,77]. However, the species remains appropriately classified as endangered, and would need to increase from a current population size to more than 2,500 mature individuals before qualifying for downlisting to vulnerable. It is also important to note that some improvements in status under criterion A may simply reflect slower rates of decline, rather than recovery. For example, Fiery-throated Fruiteater (*Pipreola chlorolepidota)* which was downlisted from vulnerable to near threatened owing to a decline in the rate of habitat loss (Table 2). Given the moderate sensitivity of the RLI, there is a strong case for complementing its application at the national scale (requiring periodic repeated red list reassessments of species) with the development of population abundance indices (like the Wild Bird Index; [78]) based on systematic population monitoring schemes (as recently established in some African countries; [79]).

## Threats to birds in Colombia

Overall, the status of birds in Colombia has deteriorated during 2002–2016, with four times as many species declining in condition sufficiently to qualify for uplisting than the number of species improving and qualifying for downlisting. As it is the case globally, loss or degradation of habitat is the main threat to Colombian birds according to our results. Illegal economic activities are the main drivers of habitat loss and degradation within the country [43,80], exceeding the impact of legal agriculture, timber extraction and mining, at least for threatened species. Illicit coca crops are the single most important driver of habitat loss, affecting eight of 13 uplisted species (Table 2). This is particularly severe in the south of Pacific and south Andean regions on the border with Ecuador, but also in the Catatumbo subregion on the border with Venezuela [81]. Illegal logging and illegal mining are also important causes of habitat loss in some regions, affecting two uplisted species. Illegal logging accounts for 42% of logging in Colombia and occurs mainly in Darién (Pacific Region) and Amazon [82]. Illegal mining is also a significant cause of habitat loss and degradation in several areas. For example, 99% of the mines of Chocó (Pacific region) operate illegally, according to the 2011 Departmental Mining Census conducted by the Ministry of Mines and Energy [83].

Illegal trafficking of drugs, arms, timber, gold, and coltan (a valuable mineral), respond to international markets. They have fueled armed and political conflict, and benefit from the

permanence of such conflicts (e.g. [84,85]). In the particular context of Colombia, armed conflict has left a long history of environmental degradation from illicit coca crops, illegal gold mining, etc., and a more ephemeral phenomenon of illegal poppy crops [86]. The spatial distribution of coca cultivation has been very dynamic in the country, especially since the beginning of the 21st century. Between 2001 and 2006, Guaviare and Putumayo in the Amazon basin were the two primary producer regions [57]. Eradication efforts with aerial sprays of glyphosate (a potent herbicide) in the Amazonian region promoted the displacement of illicit crops to other areas with low state presence and more vulnerable ecosystems [57], such as parts of Nariño department (Pacific) which previously lacked coca plantations, consequently resulting in increased deforestation in southern Chocó region [57,87,88]. This has been one of the most dramatic spatial trends, at least noted by the academic community studying illegal drugs dynamics (L. Dávalos, com. pers). Thus, coca crops strongly affected ecosystems of the Pacific region between 2001 and 2008 [89]. Threats whose driving forces derive from illegal transnational activities require international cooperation to address them.

Fire, affecting almost the entire Orinoquia, and agricultural expansion, including for cattle ranching, affected two uplisted species, both in Orinoquia: Bearded Tachuri (*Polystictus pectoralis*) and Orinoco Goose (*Oressochen jubatus*). The former depends on natural grasslands and is affected by an increase in the frequency of fires for livestock grazing, and more locally by habitat transformation for agriculture. The subspecies *P. pectoralis bogotensis* has already been driven extinct by habitat transformation in the Eastern Andes and the dry valleys in the west of the country [42]. Orinoco Goose depends on seasonally flooded savannas, wetlands, and forest for feeding and nesting, habitats that are all being affected by the expansion of agriculture and intensive cattle-raising [90]. Habitat loss and degradation are also important threats for freshwater species, driven by artificial drainage of flooded savannas for crops [90] and by hydrology and water quality alteration by agriculture [91,92]. Freshwater species are also affected by eutrophication and invasive species such as introduced trout (*Oncorhynchus mykiss*). These factors were important in the loss of the only extinct species in Colombia, the endemic Colombian Grebe (*Podiceps andinus*), with the Northern Silvery Grebe (*Podiceps juninensis*) currently following a very similar trajectory at a national scale [93,94]. Finally, logging and destruction of mangroves affect one species (Guayaquil woodpecker *Campephilus gayaquilensis*). Colombia has experienced the second highest loss of mangrove cover in the Eastern Tropical Pacific and has the lowest protected area coverage of mangroves in the region (23.7%; [95]).

Hunting is the second most important threat, affecting seven of 13 uplisted species. This activity particularly affects cracids (guans and curassows, driving the low RLI values for gamebirds), freshwater species, large raptors and some other species (see Table 2). Although hunting for sport is illegal in the country, subsistence hunting is a sensitive issue because it involves traditional practices and the food security of indigenous communities [96,97]. It is important to have a better understanding of motivations of illegal hunters in order to generate actions that are sensitive to socio-political and economic contexts [98]. Hunting typically operates in synergy with others threats, such as illegal mining and timber extraction, because miners and loggers operating in remote areas often hunt for food. Invasive animals are a more local, but significant threat, affecting three uplisted species, and involving both alien species (e.g. dogs and cats) as well as native species such as the Shiny Cowbird (*Molothrus bonariensis*) that has recently expanded its distribution within the country following clearance and degradation of native forests (e.g. [99]).

## Implications for conservation

Our results support the use of the Red list index as an indicator of trends in risk extinction for monitoring biodiversity. This is particularly important in a country like Colombia, with

dynamic spatial patterns of habitat loss, and high levels of endemism and species richness. The processes that affect the future of Colombian birds include some operating at a broad scale and others at finer scales, requiring different set of management strategies and relevant actors. At the core of large-scale processes lies the interaction between armed conflict and illegal business. Armed conflict and illegal activities undermine the capacity of institutions to control the territory and to provide appropriate environmental management. Other drivers of habitat loss and threats operate at a finer scale. Focused action is required to address them and to support species recovery. Interventions such as habitat management and restoration, hunting controls, invasive species management, and environmental education are needed for the recovery of many species. Successful examples of such action led to the downlisting of the Yellow-eared Parrot (*Ognorhynchus icterotis*). In particular, species in paramo, high Andean forest, and freshwater habitats, plus mangrove specialists, threatened seabirds, and species with extremely small distributions can benefit from such local actions. Conservation actions need to take into account the interaction with these broad and finer scale processes or otherwise they are bound to fail.

Recent developments provide reasons for both hope and despair. By November 2016, a peace agreement was signed between the Colombian government and the FARC guerrilla to end 50 years of war, which raised great hopes for increased institutional and political stability, and thus institutional capacity building. The positive consequences to avifauna conservation are twofold. There is a growing enthusiasm for the observation and study of birds, as well as a sense of pride in the national media. This has taken place with an increased use of online databases that store information with huge potential for citizen science, such as eBird and Xeno-canto. These databases are improving knowledge on birds in places that were previously poorly known. There are also an increasing number of scientific expeditions to places previously unexplored that the conflict used to make too dangerous to visit, some of them involve former insurgents who are in the process of reintegration to civil society [100]. Furthermore, the absence of conflict provides a chance to strengthen bird-watching tourism benefiting local communities. Recent studies show that the potential revenues that bird-watching tourism could bring to communities in this post-conflict period are promising [101,102]. This new opportunity provided by bird-watching tourism is a good way to offer sources of income in post-conflict zones without destroying nature [101,103]. Some of these new enterprises could also involve ex-combatants as local guides, taking advantage of their local knowledge of forest, and providing further opportunities to incorporate them into the civil society [104].

However, all these new opportunities for biodiversity conservation, local development based on bird-watching tourism, and advancement in scientific knowledge coexist with more worrying trends. Since late 2016, remnant illegal armed groups have flourished in post-conflict zones, with a return to violence and an increase in homicides as the Colombian government has struggled to reassert itself in the areas formerly controlled by FARC [105,106]. From 2017 to 2018, there was a six-fold increase in fires inside protected areas formerly under FARC control [107]. Based on local history in Guaviare department (Amazon), deforestation has increased from pastures, cattle ranching and land grabbing [108]. The area of land deforested in 2017 increased by 23% compared with 2016 [109]. Two municipalities in the Amazon (San Vicente del Caguán and Cartagena del Chairá, Caquetá department) contributed a staggering 34.6% of all national deforestation in 2018 [63]. Close monitoring and rapid responses are required to address such dynamic threats. The outcome will be heavily influenced by effectiveness of efforts to stabilize this complex situation.

Such periods of social instability are well documented in post-conflict societies [110]. The rise of violence post-civil war may undermine the legitimacy of the new peace order [111], which may have important environmental consequences. For instance, in countries such as

Nepal, Sri Lanka, Ivory Coast and Peru there was on average a 68% increase of annual forest loss in the five years following the end of armed conflict [110], contrasting with a worldwide annual loss of 7.2% of forest [73]. Inappropriate governance and institutional arrangements have been identified as the key deforestation driver during the transition period after a war. As a whole, the drivers of deforestation after the end of armed conflicts are a complex mix of social, political, institutional and governance related issues [110].

Finally, these new conservation opportunities should be a strong motivation for environmental institutions, scientists, politicians, and civil society to take advantage of the fact that the overall risk of extinction of birds in Colombia is still low and has been rather stable. Effective actions are urgently needed while there is still the opportunity to prevent extinctions and safeguard species, particularly those in higher risk categories.

## Supporting information

**S1 Table. Disaggregated Red List Index values for 2002 and 2016 for different regions, ecosystems and species groups.** There were no species that underwent genuine changes in status in lowland dry forest, paramo, coastal waters, Caribbean Coast and Ocean, Darién, SNSM, San Andrés and Providencia, nor among nocturnal raptors.
(PDF)

**S2 Table. Data used to calculate disaggregated red list indices.** Species are classified by regions, ecosystems and groups of conservation concern. The latter include Trochilidae (hummingbirds), Tytonidae and Strigidae (nocturnal raptors), Psittacidae (parrots) and Suboscines and Oscines passerines. Abbreviations: L. rain-forest = lowland rainforest; Sub-A. forest = sub-Andean forest; High A. forest = high-Andean forest; L. dry forest = lowland dry forest; Fwt. = Freshwater ecosystems; C.wt = coastal and pelagic waters. P. Ocean = Pacific Ocean; C. Sea = Caribbean Sea; SNSM = Sierra Nevada de Santa Marta; Sa&Pr = San Andrés & Providencia islands. T. forest insectiv. = terrestrial forest insectivores; Large Frugiv. = large frugivores. Taxonomy following South American Classification Committee (SACC), 2018. National and global categories according to Renjifo et al. 2002, Renjifo et al. 2016 and BirdLife International 2018 respectively.
(PDF)

## Acknowledgments

First of all, we would like to thank the hundreds of coauthors, individuals, and institutions who contributed to the red data books of Colombia and BirdLife's global IUCN Red List assessments. We are grateful with Nicholas Bayly and Felipe Estela for sharing their knowledge about migratory, and marine species and their distributions, and with Juan David Amaya for encouraging the work for a RLI for Colombia. We also thank Nicolás Giraldo for his help in compiling data, and Rob Martin for sharing with us BirdLife International´s digital checklist of the birds of the world. We also thank Angela María Forero for preparing a map of the regions in Colombia. We are especially grateful to Liliana Dávalos, her review and comments contributed to the improvement of the manuscript. We appreciate comments by Karolina Fierro and Carlos Ruiz (from Asociación Calidris) which helped to highlights some aspects in this paper, as well as the useful comments by two anonymous reviewers whose contributions improved content of this paper.

## Author Contributions

**Conceptualization:** Luis Miguel Renjifo, Angela María Amaya-Villarreal, Stuart H. M. Butchart.

**Data curation:** Luis Miguel Renjifo, Angela María Amaya-Villarreal.

**Formal analysis:** Luis Miguel Renjifo, Angela María Amaya-Villarreal, Stuart H. M. Butchart.

**Funding acquisition:** Luis Miguel Renjifo.

**Investigation:** Luis Miguel Renjifo, Angela María Amaya-Villarreal.

**Methodology:** Luis Miguel Renjifo, Angela María Amaya-Villarreal, Stuart H. M. Butchart.

**Project administration:** Luis Miguel Renjifo.

**Supervision:** Luis Miguel Renjifo.

**Visualization:** Angela María Amaya-Villarreal.

**Writing – original draft:** Angela María Amaya-Villarreal.

**Writing – review & editing:** Luis Miguel Renjifo, Angela María Amaya-Villarreal, Stuart H. M. Butchart.

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
