## [Decision Letter · Decision Letter 0]

25 Oct 2019

PONE-D-19-20151

Tracking extinction risk trends and patterns in a mega-diverse country: A Red List Index for birds in Colombia

PLOS ONE

Dear Dr. Renjifo,

Thank you for submitting your manuscript to PLOS ONE. After careful consideration, we feel that it has merit but does not fully meet PLOS ONE’s publication criteria as it currently stands. Therefore, we invite you to submit a revised version of the manuscript that addresses the points raised during the review process.

This ms is somewhere between a minor and major revision. There are two major changes needed, and a few smaller ones. First, as suggested by reviewer #2, a more comprehensive statistical analysis to substantiate the major claims would greatly strengthen the conclusions. Second, a supplementary table showing all species would seem like a very useful addition that should not be too terribly difficult to assemble. Minor points include the claim of where the highest species richness occurs (Colombia or Brazil?). Certainly both have very high richness! Perhaps a rewording to indicate this with a citation to back up the claim would be useful. Please consider the other suggestions provided by the reviewers. 

We would appreciate receiving your revised manuscript by Dec 09 2019 11:59PM. To enhance the reproducibility of your results, we recommend that if applicable you deposit your laboratory protocols in protocols.io, where a protocol can be assigned its own identifier (DOI) such that it can be cited independently in the future. For instructions see: http://journals.plos.org/plosone/s/submission-guidelines#loc-laboratory-protocols

We look forward to receiving your revised manuscript.

Kind regards,

Tim A. Mousseau

Academic Editor

PLOS ONE

**Journal Requirements:**

**Comments to the Author**

1. Is the manuscript technically sound, and do the data support the conclusions?

Reviewer #1: Yes

Reviewer #2: Yes

2. Has the statistical analysis been performed appropriately and rigorously? 

Reviewer #1: Yes

Reviewer #2: No

3. Have the authors made all data underlying the findings in their manuscript fully available?

Reviewer #1: Yes

Reviewer #2: No

4. Is the manuscript presented in an intelligible fashion and written in standard English?

Reviewer #1: Yes

Reviewer #2: Yes

5. Review Comments to the Author

Reviewer #1: Dear authors

Brazil has the richest avifauna in the world (see Piacentini et al. 2015), with 1.919 species. Please change this information throughout the manuscript. Please change the Y-axis of the graphs - it is more intuitive for the reader to perceive the increase of the extinction risk looking up, to the tip of the arrow. As shown in the ms it is not very intuitive and the output of the graphs must be changed. The results are well presented, clear and concise. However, the discussion should be improved and supported by more data or references (e.g. There are no protected areas within the Darién Highlands in Colombia, and there 270 is very limited protection within San Andrés and Providencia, lacking a reference or vague information - what's very limited?). Another example: "again suggesting that, on average, the level of threat to birds in Colombia is worsening less rapidly than at a global scale.", needs a reference or a more precise comparison. The discussion needs to be improved. The section "Threats to birds in Colombia" is very important, and should be improved with scientific data - some parts of the ms are lacking data which could be obtained quite easily (e.g. Agricultural expansion, including for cattle ranching, affected two uplisted species, both in 347 Orinoquia: Bearded Tachuri (Polystictus pectoralis) and Orinoco Goose (Oressochen jubatus).) How? These species are extremely distinct in habitat requirements and behavior and other natural history traits, and, despite briefly described in the table, need to be better detailed. Orinoco Goose is suffering from the expansion of rice fields? What's the real threat for this species?

The ms is well-written, lacking few adjustments to be published.

Reviewer #2: The paper calculated the Red List Index for Colombian birds, comparing the values for 2002 and 2016. It is an interesting contribution and makes good use of a complete database previously gathered by the authors. They go onto explaining reasons for up or down-listing the 16 species that changed category, and also analyze the changes for different regions, ecosystems, and species groups. Thankfully, the changes in RLI are very minor.

The paper is generally well written and enjoyable to read. Presents the information needed and discusses it using other literature and up-to-date data on deforestation, armed conflict, hunting, etc.

I find one important thing missing in the paper. There should be a supplementary table with all 1718 species that the authors analyzed showing each species, its current category, the category at 2002, the global category, the region, the ecosystem, and the species group to which it was assigned. This is needed for readers to see the transparency of the study and be able to replicate it.

There is another addition that would make the paper a lot stronger, and it is the reason why I suggest major revisions. You discuss the importance of some groups in driving the national RLI, but you have no statistical test to back this up. I suggest adding group testing statistical tests, like ANOVA for each of your categories, to see which one is driving the index (statistically). This would really show which group of species, ecosystem, or region is driving the national total. Once this is incorporated to the methods, results, and discussion, the paper will be ready for a second revision and then publication.

Here are some detailed comments:

Line 28: it would be useful to explain in a short way what a higher index means. Higher index, more threat? This is key because then you go on to explaining your results and the reader might not know about the RLI.

Line 35: maybe it should read “…are the most threatened”, or “…For x, y, ans z the threat status has worsened/the extinction risk increased”

Line 36: invasive animals, or species? Invasive plants are often also involved in declining populations of birds

Line 54: check the use of the word “it” in “it consumes”

Lines 77-80: check sentence and rephrase to make more clear

Line 83: a “sampled approach”..what does that mean? Using field data? Ground trothing? Please clarify

Line 85-86: change to “where conservation resources are also typically allocated”

Line 104-106: this reads more like a discussion that part of the introduction. Leave it out here and include it in the discussion of your results.

Line 118: it is not clear which are “these sources”. Do you mean the extinction assessments mentioned above? Make this clear.

Line 145-151: a map delineating these regions would be a really nice addition to the paper. I suggest including it.

Line 158-159: why was the Darien treated differently? Explain and maybe include a source

Lines 160-170: this would be easier to read in a table. Also I strongly advise including a supplementary table with how you classified each species in each region, ecosystem, and species group so readers can track what is happening with each species, and so they can replicate the study.

Line 204-206: Does this mean you can only use this 18% in your analyses. You should clarify that.

Lines 201-222: Only cite Table 2 once. Currently you cite it like 4 times in the paragraph.

Lines 229-237: I would suggest using a statistical test that compares differences between groups, to see if different ecosystems, regions, or species groups had significant impact on the overall change of RLI. Maybe an ANOVA.

Table 2: Orinoco Goose: you mention forest loss as important for this species, it probably is due to their nesting grounds, but not so much their every-day life. Please specify that forest is needed for nesting, I think this would help the reader understand the species situation better.

In Black Inca: should be “mature” not “madure”

In Micrastur: “over the next…” and “illegal coca crops”

In Pipreola:” moved to other regions”

In Cistothorus: rephrase last sentence of the explanation. It should read better.

Lines 252-255: this claim could use a reference to back it up

Lines 259-260: I would describe what happens in the Andes as deforestation, and not logging. Logging can be low impact, or selected (like they do in the Amazon), but the Andes have mostly been deforested for agriculture, pasture, or urbanization.

Lines 261-262: Again, a statistical analysis testing the importance of groups is needed, and one to highlight which group has the strongest impact.

Lines 263-266: As you mentioned in the abstract, the peace process has played its role. Most of the Amazon deforestation is very recent and has happened after the signing of the peace treaty. Check out information by the project MAAP (https://maaproject.org/en/ ) to get some updated deforestation statistics. Or IDEAM statistics would also be good to cite here.

Line 279: “less rapidly” should be changed to “slower”, “at a slower pace”, or something similar.

Line 318: need a comma before “at least for threatened species”

Lines 348-350: are oil palm plantations in savannas really that large to have a noticeable impact? Or are you referring more to the potential impact they could have. As I understand it, these plantations are very few and far between still (in savannas).

Lines 363-364: fins a reference to back up this statement. There are social scientists studying subsistence hunting and papers are out there.

References: check ALL your references for style. I found some mistakes but did not check all the references. Please make sure they are all correct before resubmitting your manuscript.

6. PLOS authors have the option to publish the peer review history of their article (what does this mean?). If published, this will include your full peer review and any attached files.

Reviewer #1: No

Reviewer #2: No

---

## [Author Response · Author response to Decision Letter 0]

9 Dec 2019

Thank you for your kind response and time dedicated to our submission. Following your indications, please find enclosed the revised version of our original manuscript PONE-D-19-20151 entitled “Tracking extinction risk trends and patterns in a mega-diverse country: A Red List Index for birds in Colombia” which we submitted to be considered for publication in PLoS ONE, as a research article. 

We are grateful for the relevant contributions by you and the reviewers. Undoubtedly most of their suggestions allowed to improve our article. In this letter, I first will answer your comments and then, one by one comments of the reviewers. 

We considered very carefully the point about “a more comprehensive statistical analysis to substantiate the major claims would greatly strengthen the conclusions”. The central request by the reviewer is to partition the proportion of variance due to different factors. However, our data are not continuous, they are categorical (regions, ecosystems and species groups of conservation concern). Further, our data are counts (numbers of species in each region, ecosystem or group, and in each IUCN category, etc.), thus they don´t have variances, and they are not completely independent from each other. For instance, species can be found in one or several regions. Perhaps more importantly, risk assessments were conducted at a national level lumping all regions together. This means that the data are unsuitable for an analysis to partition the proportion of variation due to different factors. However, in response to this important suggestion, we identified statements in the paper that required further statistical analysis to strengthen them. We performed additional contingency table tests, following the specific suggestion by the reviewer in order to determine statistically which regions are driving the Red List Index trends.

We agree with the suggestions by both reviewers to include a supplementary table showing the underlying data that we used to calculate the disaggregates indexes, with all species with categories of risk (2002, 2016 and global) and classified by region, ecosystem, and group of conservation concern. We added this as Supplementary material (S1 and S2).

Finally, we provided references to support our statement of Colombia having the most species rich avifauna in the world. Further, we checked the reference Piacentini et al. 2015 regarding Brazil as that county with the richest avifauna, but we found that Piacentini et al. 2015 explain why Brazil has the second or third richness of bird species, after Colombia and perhaps Peru. 

Please find below our responses to each comment of the reviewers. We look forward to your final decision on our manuscript. I would like to highlight that line numbers correspond to “Revised manuscript with track changes”.

Best regards,

Luis Miguel Renjifo, PhD.

School of Environmental and Rural Studies

Pontificia Universidad Javeriana

Carrera 7ª N° 40-62

Edificio Emilio Arango Piso 4

Bogotá, Colombia

Tel: (57-1)3208320 Ext 3438

lmrenjifo@javeriana.edu.co

Reviewer #1: 

First of all, we would like to thank your detailed and helpful comments. They have contributed to improve our paper in an important way. 

Dear authors Brazil has the richest avifauna in the world (see Piacentini et al. 2015), with 1.919 species. Please change this information throughout the manuscript. 

Response: Differences in species numbers between different sources are the result of both species diversity and the taxonomic philosophy underlying each species list. However, comparing works using the same taxonomic approach (BirdLife International http://www.birdlife.org/datazone/country/colombia and South America Classification Committee http://www.museum.lsu.edu/~Remsen/SACCBaseline.htm), Brazil comes in third place at a global scale after Colombia and Peru. In fact, Piacentini et al. 2015 state that: “Forming almost half of the “Bird Continent” of South America, Brazil vies for the title of the country with the richest avifauna along with Colombia and Peru (the latest statistics put it in second, 

after Colombia; Remsen et al. 2015).” Appropriate references have therefore been inserted in the text to justify this statement (see lines 122-124). 

Please change the Y-axis of the graphs - it is more intuitive for the reader to perceive the increase of the extinction risk looking up, to the tip of the arrow. As shown in the ms it is not very intuitive and the output of the graphs must be changed. 

Response: The Red List Index is widely used and recognized now, being published in dozens of papers, used by the CBD to track progress to the Aichi targets (https://www.cbd.int/gbo4/), adopted by the United Nations to track progress to the Sustainable Development Goals (https://unstats.un.org/sdgs/indicators/database/), and included in many assessments and reports and policy processes (e.g. the Global Assessment of the Intergovernmental Science-Policy Platform on Biodiversity and Ecosystem Services). The index was specifically formulated so that the graph goes down as things get worse, to be consistent with most other biodiversity indicators (eg the Living Planet Index, Wild Bird Index, Biodiversity Intactness Index etc). We therefore prefer to retain its current format for consistency and to avoid confusion. However, we added new text to explain this specifically. (See lines 181-183).

The results are well presented, clear and concise. However, the discussion should be improved and supported by more data or references (e.g. There are no protected areas within the Darién Highlands in Colombia, and there is very limited protection within San Andrés and Providencia, lacking a reference or vague information - what's very limited?). 

Response: We have edited the sentence to improve clarity and added a reference (see lines 397-399): “Los Katíos National Park covers only part of the Darien lowlands, but no highland species are protected within it, and there is only one small terrestrial protected area on each of San Andrés and Providencia islands (RUNAP 2019).”

Another example: "again suggesting that, on average, the level of threat to birds in Colombia is worsening less rapidly than at a global scale.", needs a reference or a more precise comparison. 

Response: We have provided the precise numeric values for decline rates and inserted the appropriate reference in line 409 for the global index. 

The discussion needs to be improved. The section "Threats to birds in Colombia" is very important, and should be improved with scientific data - some parts of the ms are lacking data which could be obtained quite easily (e.g. Agricultural expansion, including for cattle ranching, affected two uplisted species, both in Orinoquia: Bearded Tachuri (Polystictus pectoralis) and Orinoco Goose (Oressochen jubatus).) How? These species are extremely distinct in habitat requirements and behavior and other natural history traits, and, despite briefly described in the table, need to be better detailed. Orinoco Goose is suffering from the expansion of rice fields? What's the real threat for this species?

Response: We have included further detail on factors affecting both species see lines 494-502: Fire, affecting almost the entire Orinoquia, and agricultural expansion, including for cattle ranching, affected two uplisted species, both in Orinoquia: Bearded Tachuri (Polystictus pectoralis) and Orinoco Goose (Oressochen jubatus). The former depends on natural grasslands and is affected by an increase in the frequency of fires for livestock grazing, and more locally by habitat transformation for agriculture. The subspecies P. pectoralis bogotensis has already been driven extinct by habitat transformation in the Eastern Andes and the dry valleys in the west of the country [42]. Orinoco Goose depends on seasonally flooded savannas, wetlands, and forest for feeding and nesting, habitats that are all being affected by the expansion of agriculture and intensive cattle-raising [90].

The ms is well-written, lacking few adjustments to be published.

Reviewer #2: 

First of all, the authors would like to thank your detailed and helpful comments. They have contributed to improve our paper in an important way. 

I find one important thing missing in the paper. There should be a supplementary table with all 1718 species that the authors analyzed showing each species, its current category, the category at 2002, the global category, the region, the ecosystem, and the species group to which it was assigned. This is needed for readers to see the transparency of the study and be able to replicate it.

Response: Absolutely. A supplementary table has been included as S2.  

There is another addition that would make the paper a lot stronger, and it is the reason why I suggest major revisions. You discuss the importance of some groups in driving the national RLI, but you have no statistical test to back this up. I suggest adding group testing statistical tests, like ANOVA for each of your categories, to see which one is driving the index (statistically). This would really show which group of species, ecosystem, or region is driving the national total. Once this is incorporated to the methods, results, and discussion, the paper will be ready for a second revision and then publication.

Response: this is a good point but it is important to consider in relation to the nature of the data. The key point is to partition the proportion of variance due to different factors. However, our data are categorical and not continuous (regions, ecosystems and species groups of conservation concern). Further, our data are counts (numbers of species in each region, ecosystem or group, and in each IUCN category, etc.), and they are not entirely independent from each other. For instance, species can be found in one or several regions. Perhaps more importantly, risk assessments were conducted at a national level, lumping all regions together. Therefore, although this sort of analysis would have been highly desirable, the data do not permit it. However, we identified statements in the paper that required further statistical analysis to justify them. We therefore performed additional contingency tables analyses, following the specific suggestion by the reviewer in order to determine which regions are statistically driving the Red List Index (RLI), and used contingency table analysis when expected numbers in the contingency tables were greater than 5 (see lines 305-309).

Here are some detailed comments:

Line 28: it would be useful to explain in a short way what a higher index means. Higher index, more threat? This is key because then you go on to explaining your results and the reader might not know about the RLI.

Response: Good point. We have added text to clarify this. See the current lines 28-29.

Line 35: maybe it should read “…are the most threatened”, or “…For x, y, ans z the threat status has worsened/the extinction risk increased”

Response: Done. See the current line 36.

Line 36: invasive animals, or species? Invasive plants are often also involved in declining populations of birds.

Response: We meant animals and have edited the text to clarify this. See the current line 38.

Line 54: check the use of the word “it” in “it consumes”

Response: Done. See the current line 59.

Lines 77-80: check sentence and rephrase to make more clear

Response: We have reworded this sentence. See the current lines 85-87.

Line 83: a “sampled approach”..what does that mean? Using field data? Ground trothing? Please clarify.

Response: We have provided additional text to clarify this, (see lines 90-93): The sampled approach involves assessing random selection of species from across the entire taxonomic group, allowing for the assessment of trends for large taxonomic groups [Baillie et al. 2008].

Line 85-86: change to “where conservation resources are also typically allocated”

Response: Done. See the current line 95-96.

Line 104-106: this reads more like a discussion that part of the introduction. Leave it out here and include it in the discussion of your results.

Response: We prefer to keep this sentence here to give a context to our study in terms of its relevance as well as previous studies leading to this paper.

Line 118: it is not clear which are “these sources”. Do you mean the extinction assessments mentioned above? Make this clear.

Response: We have edited to text to make this clear. See current line 141.

Line 145-151: a map delineating these regions would be a really nice addition to the paper. I suggest including it.

Response: We agree. Please see the current line 201 and the new figure 1.

Line 158-159: why was the Darien treated differently? Explain and maybe include a source. 

Response: We have now provided an explanation: This is because on small isolated mountains the upper limit of lowland rain forest is lower than on major ranges due to the “Massenerhebung” effect (Grubb 1971). See current lines 208-210.

Lines 160-170: this would be easier to read in a table. Also I strongly advise including a supplementary table with how you classified each species in each region, ecosystem, and species group so readers can track what is happening with each species, and so they can replicate the study.

Response: This point is addressed with the inclusion of the new supplementary table (S2), as it includes the whole database with the classifications of each species in each region, group and other classes.

Line 204-206: Does this mean you can only use this 18% in your analyses. You should clarify that.

Response: The calculation of the RLI uses the complete list of species and their current red list categories, but trends since the first time-point are driven only by the changes in categories for the 18% of species for which these were genuine. We consider have added text to clarify this.:” These category changes drive trends in the RLI, although the full set of species and their current red list categories are used to determine the 2016 RLI value.” See lines 271-273.

Lines 201-222: Only cite Table 2 once. Currently you cite it like 4 times in the paragraph.

Response: Done. See page 10.

Lines 229-237: I would suggest using a statistical test that compares differences between groups, to see if different ecosystems, regions, or species groups had significant impact on the overall change of RLI. Maybe an ANOVA.

Response: Please see second response to reviewer 2. 

Table 2: Orinoco Goose: you mention forest loss as important for this species, it probably is due to their nesting grounds, but not so much their every-day life. Please specify that forest is needed for nesting, I think this would help the reader understand the species situation better.

Response: Yes, we agree. We specify in table 2 (see the beginning of the table at line 328), that forest is needed for nesting.

In Black Inca: should be “mature” not “madure”

In Micrastur: “over the next…” and “illegal coca crops”

In Pipreola:” moved to other regions”

In Cistothorus: rephrase last sentence of the explanation. It should read better.

Response: all these minor changes were done in table 2.

Lines 252-255: this claim could use a reference to back it up. 

Response: This is our finding and interpretation. We have edited the wording to clarify this (see current lines 368-369).

Lines 259-260: I would describe what happens in the Andes as deforestation, and not logging. Logging can be low impact, or selected (like they do in the Amazon), but the Andes have mostly been deforested for agriculture, pasture, or urbanization.

Response: we changed forest logging to “deforestation”. See current line 373.

Lines 261-262: Again, a statistical analysis testing the importance of groups is needed, and one to highlight which group has the strongest impact. 

Response: We agree, so we performed new G-tests (see lines 231-233 also 306-309.

Lines 263-266: As you mentioned in the abstract, the peace process has played its role. Most of the Amazon deforestation is very recent and has happened after the signing of the peace treaty. Check out information by the project MAAP (https://maaproject.org/en/ ) to get some updated deforestation statistics. Or IDEAM statistics would also be good to cite here.

Response: Thank you for the suggestion. We cited both, MAAP and Ideam 2019 (See current line 382).

Line 279: “less rapidly” should be changed to “slower”, “at a slower pace”, or something similar.

Response: Done. See line 411

Line 318: need a comma before “at least for threatened species”

Response: Done. See line 458

Lines 348-350: are oil palm plantations in savannas really that large to have a noticeable impact? Or are you referring more to the potential impact they could have. As I understand it, these plantations are very few and far between still (in savannas).

Response: This needed to be written more clearly. We edited this, and removed a reference oil palm: “Fire, affecting almost the entire Orinoquia, and agricultural expansion, including for cattle ranching, affected two uplisted species, both in Orinoquia: Bearded Tachuri (Polystictus pectoralis) and Orinoco Goose (Oressochen jubatus). The former depends on natural grasslands and is affected by an increase in the frequency of fires for livestock grazing, and more locally by habitat transformation for agriculture. The subspecies P. pectoralis bogotensis has already been driven extinct in by the habitat transformation in the Eastern Andes and the dry valleys in the west of the country [42]. Orinoco Goose depends on seasonally flooded savannas, wetlands, and forest for feeding and nesting, habitats that are all being affected by the expansion of agriculture and intensive cattle-raising [90].” Lines 495-503.

Lines 363-364: fins a reference to back up this statement. There are social scientists studying subsistence hunting and papers are out there.

Response: We added two appropiate references regarding this point (Robinson and Redford 1991, Sarti et al 2015). See current line 528.

References: check ALL your references for style. I found some mistakes but did not check all the references. Please make sure they are all correct before resubmitting your manuscript.

 Response: We checked again all references to correct a few mistakes.

---

## [Editor Report · Decision Letter 1]

18 Dec 2019

Tracking extinction risk trends and patterns in a mega-diverse country: A Red List Index for birds in Colombia

PONE-D-19-20151R1

Dear Dr. Renjifo,

We are pleased to inform you that your manuscript has been judged scientifically suitable for publication and will be formally accepted for publication once it complies with all outstanding technical requirements.

With kind regards,

Tim A. Mousseau

Academic Editor

PLOS ONE

---

## [Editor Report · Acceptance letter]

20 Dec 2019

PONE-D-19-20151R1 

Tracking extinction risk trends and patterns in a mega-diverse country: A Red List Index for birds in Colombia 

Dear Dr. Renjifo:

I am pleased to inform you that your manuscript has been deemed suitable for publication in PLOS ONE. Congratulations! Your manuscript is now with our production department. 

With kind regards,

on behalf of

Dr. Tim A. Mousseau 

Academic Editor

PLOS ONE